# THE N-BODY PROBLEM: PREDICTING PARALLEL EXECUTION FROM SINGLE-PERSON EGOCENTRIC VIDEO

## ABSTRACT

Humans can intuitively parallelise complex activities, but can a model learn this from observing a single person? Given one egocentric video, we introduce the N-Body Problem: how $N$ individuals, can together perform the same set of tasks observed in this video. The goal is to maximise speed-up, but naive task allocation often violates real-world constraints, leading to physically impossible scenarios like two people using the same object or occupying the same space. To address this, we formalise the N-Body Problem and propose a suite of metrics to evaluate both performance (speed-up, task coverage) and feasibility (spatial collisions, object conflicts). We then introduce a structured prompting strategy that guides a Vision-Language Model (VLM) to reason about the 3D environment, object usage, and temporal dependencies to produce a viable parallel execution. On 100 videos from EPIC-Kitchens and HD-EPIC, our method for $N = 2$ boosts action coverage by 45% over a baseline prompt for Gemini 2.5 Pro, while simultaneously slashing collision rates by 55% and object conflicts by 45%.

## 1 INTRODUCTION

As humans, we can naturally decide when and where to split tasks so as to complete all these tasks faster, without disrupting the flow of others. This suggests that even tasks performed by a single person contain hidden opportunities for parallelisation. In unscripted egocentric videos, prior works have shown that the camera wearer often carry out multiple goals in parallel Rubinstein et al. (2001); Monsell (2003); Price et al. (2022); Shen & Elhamifar (2025). This raises an intriguing question: can we predict a parallel execution of multiple agents, such that the original work can be sped up?

We introduce and formalise **the N-Body Problem** – the novel problem of predicting a multi-agent parallel execution from a video showcasing a single-agent execution trace. Our output should imagine a highly synergetic execution which maximises the concurrent work performed by all agents. This effective execution is not merely an optimisation problem for speed. It is fundamentally constrained by the physics and logic of the real world. Any valid parallel execution must respect several categories of constraints: First, the *Spatial Constraints*. As physical entities, the $N$ agents cannot occupy the same space simultaneously. The execution must therefore aim to be collision-free. Second, the *Object Constraints*. The execution must enforce exclusive object ownership, ensuring that no two agents attempt to use the same object at the same time. Finally, the *Causality Constraints*. To be causally correct, the execution must respect when certain actions are prerequisites for others.

Our formulation is distinguished from prior works on task graphs that capture action causality Ashutosh et al. (2023); Grauman et al. (2024); Mao et al. (2023) in three principal aspects. First, these works guide one person to complete the task and do not address parallel execution. Second, we do not assume knowledge of the tasks or pre-defined action segments. Third, we introduce an explicit focus on spatial and object reasoning. Predicting a parallel execution requires a holistic understanding not only of what actions to perform, but also of where at and with-what-objects to perform them.

We evaluate the N-Body Problem on 100 long videos from two single-person datasets Damen et al. (2022); Perrett et al. (2025) (avg 25min). These datasets provide ground-truth covering actions, object tracks and camera trajectories. Such annotations are related to the constraints in the N-Body Problem, we thus propose a set of metrics suitable for evaluating the outcome of parallelisation.

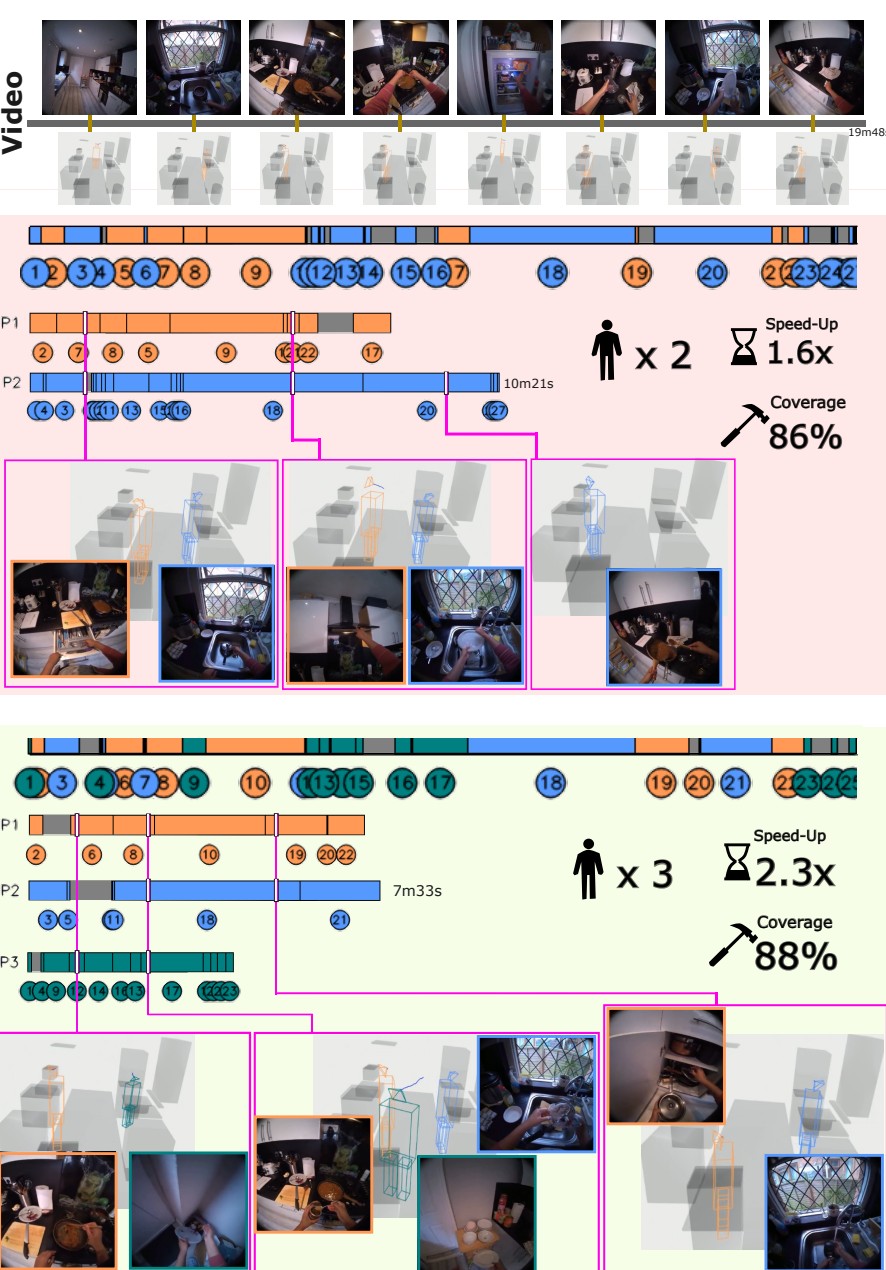

Figure 1: **See Video Supp: Top.** Our input is single-person egocentric video. The camera wearer (shown also in orange in 3D) performs a combination of cooking, washing up and ordering. **Middle.** 2-Body Parallel execution. Our method achieves a 1.6x speed-up (from 19.8min to 10.4min) with 86% coverage. We show coloured segments (orange/blue) along with 3D representation of where the 2-agents are and their camera views (colour-bordered keyframes). P1 is mostly left to do the cooking, while P2 is washing up and ordering. **Bottom.** 3-Body parallel execution achieving further 2.3x speed-up to 7.5min. P2 is washing up while P3 is drying and storing things away.

We use state-of-the-art vision-language model Gemini 2.5 Pro Comanici et al. (2025) to predict the parallel execution, given that VLMs have recently shown remarkable capabilities in understanding video, semantics and objects. We propose an elaborate prompt to allow VLMs to reason about the 3D environment, through ground-plane zones, object usage and task dependencies. This allows the VLM to better capture the constraints for a viable parallel execution. Using proposed metrics, we show how Gemini can be controlled to trade-off between goals and constraints. Fig 1 shows an output of our method, where we predict a 2-Body and 3-Body executions from the same video.

To summarise, in this paper we:

- Formulate the N-Body Problem – parallelising a multi-agent execution from a single-agent trace.
- Propose an evaluation metrics suite of parallel execution using publicly available annotations.
- Design a complex prompt that allows state-of-the-art VLM to act as a reasoning model, to solve the N-Body Problem, maximising goals and respecting spatial, object and causal constraints.
- Evaluate on 100 long videos from two single-person datasets Damen et al. (2022); Perrett et al. (2025) for both 2-Body (N=2) and 3-Body (N=3) execution.
- Demonstrate the difficulty of this problem, where naive assignment and optimisation scheduling produce significant collision rate (25.9% and 39.7%) between agents, and open-weight VLM QWEN fails to speed-up. While Gemini 2.5 Pro with base prompt performs poorly, it can be guided to improve goals and respect constraints, improving action coverage to 91.3% and reducing collision rate and object conflict rate – to 7.7% and 0.64% on HD-EPIC, respectively.

## 2 PROBLEM FORMULATION

While the general gist of acquiring help from others during everyday operation is intuitive, formulating the problem as a representation mapping from one video to $N$ parallel executions needs significant attention. We first introduce the input and output representations for the N-Body Problem in Sec 2.1. We then separate the objectives into goals (Sec 2.2) and constraints (Sec 2.3). These identify what needs to be optimised (i.e. maximised) which we note as goals, versus what makes a parallel execution incorrect or invalid, which we refer to as constraints.

### 2.1 INPUT AND OUTPUT REPRESENTATIONS

We denote the source input video as $\mathcal{I}$, which represents a single-person execution from frame 1 to $T_{\mathcal{I}}$. The problem is to divide $\mathcal{I}$ into non-overlapping assignable segments,

$$S_{ij} = [\mathcal{I}_i, \mathcal{I}_j], \quad i < j; \quad \forall ij, i'j' \rightarrow IoU(S_{ij}, S_{i'j'}) = 0 \tag{1}$$

It is critical that these segments are non-overlapping so they can be assigned to a single agent in the N-Body Problem. Note that some parts of the original video $\mathcal{I}$ might not be assigned to any segment $S_{ij}$ and thus are accordingly discarded.

For an $N$-Way parallel execution $\mathcal{P} = \{\mathcal{P}_1, \mathcal{P}_2, \ldots, \mathcal{P}_N\}$, we then need to assign each of these executable segments to one of the $N$ agents. Importantly, the order of the segments can be shuffled if needed. Additionally, any agent can be in an idle state as they wait for a viable task to perform. Denote $\mathcal{P}_n$ the execution assigned to the $n$-th agent, and denote $\mathcal{A}(\cdot, \cdot)$ the assignment function,

$$\mathcal{A}(S_{ij}, \mathcal{P}_n) = \tau \tag{2}$$

means that the start of the segment $S_{ij}$ is assigned to the frame $\tau$ in $\mathcal{P}_n$. As the duration of that segment remains unchanged, the task is only to assign a start time for that segment at $\mathcal{P}_n$. Consequentially, the frames $[\tau, \tau + |S_{ij}|]$ are all allocated to the segment:

$$\mathcal{A}(S_{ij}, \mathcal{P}_n) = \tau \quad \rightarrow \mathcal{P}_n[\tau, \tau + (j - i)] = S_{ij} \tag{3}$$

Additionally, a segment can only be assigned to a single agent, *i.e.*

$$\mathcal{A}(S_{ij}, \mathcal{P}_n) \neq null \rightarrow \forall m \neq n : \mathcal{A}(S_{ij}, \mathcal{P}_m) = null, \tag{4}$$

where $\mathcal{A}(S_{ij}, \mathcal{P}_n) \neq null$ indicates that the segment $S_{ij}$ is assigned to $\mathcal{P}_n$ at some frame.

Of course there are many possible ways to divide segments and also many ways to assign these segments to agents, leading to different parallel executions $\{\mathcal{P}\}$ from the same video $\mathcal{I}$. We next define the one(s) we are after in the N-Body Problem.

### 2.2 GOALS

Our objective is to generate one parallel execution $\mathcal{P}$ that is as efficient as possible while covering as much of the work completed in the source execution $\mathcal{I}$ as possible. Formally, we aim to maximise the following quantities:

First, the coverage of the parallel execution:

$$\text{Coverage} := \frac{\text{Number of assigned frames}}{T_{\mathcal{I}}} \tag{5}$$

where we calculate the total length of all assigned – hence covered – segments in the original video.

Second, the speed-up from the original video to parallel execution:

$$\text{Speed-Up} := \frac{\text{Sequential execution time}}{\text{Parallel execution time } (T_{\mathcal{P}})}, \tag{6}$$

where $T_{\mathcal{P}}$ amounts to the time required for the agent that is working the longest.

## 2.3 CONSTRAINTS

While maximising the above objectives, the generated execution must also remain physically and logically feasible. We therefore seek to measure from only the given video, and avoid (i.e. minimise) violations of four constraints – the first two are related to the space in the scene and the second two are the semantics of objects and causal actions.

First, the collision in the 3D scene between concurrent working agents. Agents should not be occupying the same space at one time. We measure conflicts in the space through collision rate:

$$\text{Collision Rate} := \frac{\text{Number of colliding frames in } \mathcal{P}}{T_{\mathcal{P}}}, \tag{7}$$

Second, the relocation overhead – i.e. "jump" distances of agents in the parallel execution. As each agent $n$ is assigned different segments, at times the end of one segment can be at a different part of the scene than the start of the next segment. This is a violation as it expects the agent to unrealistically *teleport* in space. We measure this jump conflict as:

$$\text{Spatial Jump} := \frac{1}{N} \sum_{n=1}^{N} (\text{Average Per-Segment-Boundary Jump Distance of agent } n) \tag{8}$$

Third, the conflict of accessing the same object by different agents. Denote $\mathcal{O}$ the set of objects accessed or used in the video. A conflict is measured as the duration when multiple agents access the same object $\mathcal{O}_k \in \mathcal{O}$ in the parallel execution. The object conflict rate is defined as:

$$\text{Object Conflict Rate} := \frac{\text{Number of frames where } \exists \mathcal{O}_k \in \mathcal{O} \text{ in conflict}}{T_{\mathcal{P}}} \tag{9}$$

Note that we here make the assumption that no extra objects in the scene can be used instead - e.g. if two knifes are used during the activity, we assume these are the only knives available. Given we only rely on the observations from a single video, it will be challenging to make assumptions about additional objects or their utility – e.g. a suitable alternative knife for slicing bread.

Lastly, a causal constraint is one that measures plausible parallel executions – i.e. preserve any tasks that need to occur in a particular order or need to take a specific time to complete. For example, if one puts something in the oven, it needs to stay there for the required amount of time and this cannot be made shorter. This is different from putting an object down, then putting it away because this time can be shortened. Similarly, an agent should not wash a dish before it is actually used. This requires an understanding of causality. The above requires a semantic understanding of tasks and time-sensitive activities. We measure any causality conflict rates similarly as:

$$\text{Action Conflict Rate} := \frac{\text{Number of frames where } \exists \mathcal{C}_r \in \mathcal{C} \text{ in conflict}}{T_{\mathcal{P}}} \tag{10}$$

where $\mathcal{C}_k$ is any action in the list of actions carried out. If any action is in conflict, i.e. is out of order or occurs earlier than it should in time, it is considered in conflict.

**Objectives Summary.** We have defined multiple objectives for the N-Body Problem, as no single measure can capture the quality of a parallel execution on its own. The distinction between goals and constraints is not absolute: for example, coverage could be treated as a constraint, while jump distance could also be viewed as a goal. In this work, we treat goals as objectives to maximise and constraints as objectives to avoid or minimise. The precise implementation of goals and constraints as quantitative metrics is in Sec 3.2. Importantly, multiple parallel executions $\{\mathcal{P}\}$ can achieve the same performance (speed-up, coverage, collision/conflict rates). We consider these to be *equivalent*.

## 3 EXPERIMENTS

### 3.1 DATASETS AND ANNOTATIONS

We evaluate our formulation on two egocentric datasets: HD-EPIC Perrett et al. (2025) and EPIC-KITCHENS-100 (which we refer to as EPIC in the rest of the paper) Damen et al. (2022). Both capture unscripted kitchen-based activities in home environments, with long continuous recordings that naturally contain interleaved goals such as cooking, cleaning, and storing. We randomly select 100 long videos – 80 from HD-EPIC and 20 from EPIC, covering 24 unique scenes (9 from HD-EPIC and 15 from EPIC) – with minimum duration of 10 minutes (avg video length 25.3 minutes) to form a tractable evaluation set. The longer videos are likely to be more meaningful for parallel execution as they tend to capture more underlying activities. Crucially, both datasets provide the necessary ground-truth annotations to evaluate the goals and constraints (presented in Sec 2) using a set of dedicated metrics (see Sec 3.2). We select more videos from HD-EPIC as they contain additional ground truth that helps us evaluate object conflict as we will describe below.

**Ground-Truth Camera Poses.** Both HD-EPIC and EPIC provide ground-truth camera poses annotated using multi-sensor SLAM Engel et al. (2023) or sub-sampled frames COLMAP Tschernezki et al. (2023). Camera poses provide strong signal of where the person situates in the environment, as well as their body orientations. With the person's trajectory given by the camera poses in the source video, we are able to evaluate the Collision Rate.

**Ground-Truth Actions.** Both HD-EPIC and EPIC provide dense and detailed annotations of all action segments completed by the camera wearer. The action narrations covers all meaningful parts of the video, which we will use for measuring the coverage of important works in the original video.

**Ground-Truth Object Movements.** Recently introduced HD-EPIC provides extensive manual annotations of object movements as object tracks. These are start-end segments that correspond to the duration when an object is moved by the camera-wearer along the video. These are exhaustive and form long trajectories covering frames when the object is in motion/use versus when the object is stationary. These tracks are linked to the object instances, allowing us to identify the same instance of any object throughout the video. The presence of these annotations allow us to accurately measure object conflict.

Unfortunately, both datasets do not contain any labels for causal actions (e.g. task-graphs). While other datasets note the ability to annotate task graphs (e.g. Grauman et al. (2024)) these labels are not made public. Other large-scale datasets (e.g. Grauman et al. (2022)) do not contain camera poses, object and action annotations to allow for a comprehensive evaluation of the N-Body Problem.

### 3.2 EVALUATION METRICS

We do not have any ground truth parallel executions – i.e. we only rely on single-person video inputs. We implement evaluation metrics that directly measure the goals (section 2.2) and constraints (section 2.3), using the available data annotations, as follows: frame coverage $\uparrow$, action coverage $\uparrow$, speed-up $\uparrow$, 3D spatial collision rate $\downarrow$, object conflict rate $\downarrow$ and average jump distance $\downarrow$.

The frame coverage of the parallel execution is implemented as in Equation (5):

$$\text{Frame Coverage} := \frac{\sum |S_{ij}| \iff \exists n; \mathcal{A}(S_{ij}, \mathcal{P}_n) \neq null}{T_{\mathcal{I}}}, \tag{11}$$

where $\mathcal{A}(S_{ij}, \mathcal{P}_n) \neq null$ means the segment $S_{ij}$ is assigned to the agent $n$ in the parallel execution $\mathcal{P}$. Note again that one segment can only be assigned to a single agent.

While frame coverage is one way to measure parallel execution, some frames might be excluded that do not correspond to any tasks or actions. Instead, we use the exhaustive annotations for action segments, we consider the set of all actions in the video $\mathcal{C}$ and compute the percentage of these actions that are covered by the parallel execution. We calculate the action coverage as:

$$\text{Action Coverage} = \frac{\sum\limits_{\mathcal{C}_r \in \mathcal{C}} \mathbb{1}\big[\exists n, i, j \colon A(S_{ij}, \mathcal{P}_n) \neq null \ \& \ IOU(S_{ij}, \mathcal{C}_r) \geq 0.5\,\big]}{|\mathcal{C}|}, \tag{12}$$

i.e. we calculate the temporal overlap between the action segment $C_i$ and any assigned segments in the parallel execution $\mathcal{P}_n$. As long as any agent $n$ has a segment that temporally overlaps with the

action by more than 0.5 IOU, we consider the action to be covered by this parallel execution. We use the standard threshold of temporal overlap (0.5) which is accepted in temporal action localisation. Note that this strict threshold 0.5 implies that the action cannot be covered by multiple agents as any other agent will naturally have a threshold lower than 0.5 IOU.

To disentangle the coverage from the speed-up, we calculate the speed-up metric as:

$$\text{Speed-Up} = \frac{\sum_{t=1}^{T_{\mathcal{I}}} \mathbb{1}[\, \exists n, i, j \colon A(S_{ij}, \mathcal{P}_n) \neq null \; \& \; i \leq t \leq j \,]}{T_{\mathcal{P}}}. \tag{13}$$

Note that the speed-up metric does not account for the frames that are *not* covered by the parallel execution $\mathcal{P}$. It measures the relative speed-up from sequential execution to parallel execution solely. We disentangle the two metrics (coverage and speed-up) to avoid solutions that drop the coverage to gain speed-up.

To evaluate the spatial collision between parallel agents, we use the ground-truth camera poses to represent person's trajectory in the source video. Denote $\Gamma_n(t)$ the spatial trajectory of the $n$-th agent at frame $t$, the collision rate is implemented as:

$$\text{Collision Rate} = \frac{\sum_{t=1}^{T_{\mathcal{P}}} \mathbb{1}[\, \exists n \neq n' \colon is\_collide(\Gamma_n(t), \Gamma_{n'}(t)) \,]}{T_{\mathcal{P}}}, \tag{14}$$

where $is\_collide(\cdot, \cdot)$ is a function that determines whether two different agents occupy the same space, based on their body positions and orientations in the world coordinate system; the $is\_collide(\cdot, \cdot)$ function is implemented using the average of size of human body[1].

The jump distance is implemented as:

$$\text{Jump} = \frac{1}{N} \sum_{n=1}^{N} \frac{1}{M_n} \sum_{m=1}^{M_n - 1} \|\Gamma_n(start_{m+1}) - \Gamma_n(end_m)\|, \tag{15}$$

where $M_n$ is the total number of segments assigned to the $n$-th agent, $start_{m+1}$ is the start frame of the segment $m + 1$ and $end_m$ is the end frame of the segment $m$.

To quantify object conflict, we use the object movement tracks. Denote $\mathcal{O}$ the set of objects accesssed, moved or used in the source video. A single object conflict is the duration when multiple agents move the same object $\mathcal{O}_k \in \mathcal{O}$ in the parallel execution. The object conflict rate can thus be implemented as:

$$\text{Object Conflict Rate} = \frac{\sum_{t=1}^{T_{\mathcal{P}}} \mathbb{1}[\, \exists \mathcal{O}_k \in \mathcal{O}, \exists n \neq n' \colon \mathcal{O}_{k,n}(t) \wedge \mathcal{O}_{k,n'}(t) \,]}{T_{\mathcal{P}}}, \tag{16}$$

where $\mathcal{O}_{k,n}$ is 1 if the object $\mathcal{O}_k$ is accessed by agent $n$ at frame $t$, and 0 otherwise.

Note that we also aim for semantically plausible parallel executions – i.e. ones that preserve task order where dependencies exist. As there are currently no publicly available annotations of task dependencies, we evaluate the semantic conflict of actions qualitatively.

### 3.3 MODEL SETUP

We use Gemini 2.5 Pro Comanici et al. (2025) to generate parallel execution $\mathcal{P}$ from input video $\mathcal{I}$. The input to the Gemini model are formed by two parts: our proposed text prompt and the video input $\mathcal{I}$. We use the default 1FPS frame sampling rate to fit the maximum context.

We progressively evolve more specific system prompts to improve Gemini 2.5 Pro's performance on the N-Body Problem:

- **Base Prompt**. The basic prompt where we simply inform the model about the task, without mentioning any goals and constraints.
- **+ Goals-Only**. On top of "Base", we instruct the model to maximise the goals (Coverage and Speed-Up), without mentioning any constraints.

---

[1]46 cm wide and 25 cm deep

- **+ Goals-and-Constraints**. We instruct the model to both maximise the goals (Coverage and Speed-Up) and minimise the constraints of space, object and semantics.
- **+ Spatial Prompt**. On top of "+ Goals-and-Constraints", we provide an *additional* column-separated-value (csv) file linking spatial location to time—temporal durations in the source video. We instruct the model to avoid occupying the same zone for different agents.

Regarding the final "+ Spatial Prompt", current vision–language models – including Gemini 2.5 Pro – often struggle with robust spatial understanding. In the parallel-execution problem, simply supplying the raw trajectory of the source video as additional input does not provide Gemini with sufficient guidance (see ablation in supplementary). We explore a compact and structured spatial prompt that explicitly encodes the knowledge needed for collision avoidance. We divide the 3D environment into equal-sized zones; we divide the XY-plane as the majority of movements happen within the ground plane. We then extract the duration when the person in $\mathcal{I}$ remains with one zone, producing a list of triplets of: (start-time, end-time, zone number). We then instruct Gemini 2.5 Pro to avoid assigning two parallel agents working in the same zone concurrently.

The complete prompt with spatial prompt can be found in the Appendix A. We set Gemini parameter $temperature = 0$ and $top_p = 0.2$ for more deterministic results.

**Additional Comparison Methods.** In addition to Gemini 2.5 Pro, we evaluate an open-weight model Qwen2.5-VL-72B Wang et al. (2024b) under our full prompt and 1 FPS sampling rate. We introduce a naive baseline that divides the video into equal halves and has the two agents execute their halves concurrently, without modelling dependencies. Additionally, we evaluate a simple HEFT-style list scheduler Topcuoglu et al. (2002). For the scheduler, we turn ground-truth action timestamps into assignable segments, using the privileged knowledge of start-end times of actions, and manually induce simple precedences from verb–object cues (*e.g. take → use → put, open → close*). We then apply a standard list scheduling heuristic: at each step, select the first ready task and place it at the earliest feasible time that respects object exclusivity. We further experiment removing the ground-truth action timestamps signal: we divide the input duration into 1-minute windows and concatenate all action narrations appearing within each 1-min. We then use the same heuristics of verb-object cues to establish precedences in assigning these 1-min segments to agents.

## 3.4 MAIN RESULTS

Table 1 shows the method performances on HD-EPIC. For each metric, we report the average metric over videos. Our prime set of comparative results are for $N = 2$ (2-Body Problem). We first present the naive baseline where we just assign the first half of the video to $P1$ and the second to $P2$. Results show very high collision rate and high object collision.

Gemini 2.5 Pro with base prompt produces good speed-up (1.58x), but considerably under-covers in frames and actions (62.9% action coverage) and produces a high collision rate (17.2%). Encoding goals in the prompt (+ Goals-only) achieves the highest speed-up and improves coverage. Encoding constraints drops the speed-up but fails to reduce the spatial collision rate. Spatial prompting (**ours**) reduces the collision rate by a large margin, increases the coverage but has a slightly lower speed-up. Table 2 shows similar trend on EPIC evaluation videos. While table 1 and 2 only present averaged results across the videos, figure 2 shows the metrics distributions for Gemini2.5 as prompts is expanded. Introducing goals significantly improves the coverage. The spatial prompt consistently lowers the collision rate.

Both the naive baseline and HEFT scheduler produce significantly high collision rates, and removing the privileged ground-truth timestamps from HEFT produces a higher object conflict rate. Open-weight VLM Qwen2.5-VL-72B fails to produce any output for 42 out of the 100 videos. On the 58-video subset, it performs poorly in coverage even though it avoids collision – this model is incapable of solving this problem.

Table 1 and Table 2 also show the comparison results between 2-body and 3-body outputs from our proposed prompt. When scaling the parallelisation from 2-body to 3-body, we observe a predictable increase in both collision and object conflict rates, while with better speed-up at the same time ($1.40 \rightarrow 1.51$ on HD-EPIC, vs $1.35 \rightarrow 1.64$ on EPIC). Qualitative samples for both $N = 2$ and $N = 3$ are shown in fig. 3.

| | Method | Coverage (%)↑ | Action Cov. (%) ↑ | Speed-Up ↑ | Coll. Rate (%) ↓ | Avg. Jump (m) ↓ | OCR (%) ↓ |
|---|---|---|---|---|---|---|---|
| 2-Body Problem | Naive Half-Half | 100.0 | 100.0 | 2.00 | 25.9 | 0.00 | 3.61 |
| | HEFT 1-min | 99.9 | 100.0 | 1.93 | 39.7 | 0.65 | 2.53 |
| | HEFT GT start-end[†] | 77.6 | 99.2 | 1.82 | 59.1 | 0.28 | 0.03 |
| | Qwen2.5-VL-72B* | 63.9 | 63.4 | 0.89 | 0.2 | 0.19 | 0.00 |
| | Gemini2.5 (Base Prompt) | 61.4 | 62.9 | 1.58 | 17.2 | 0.53 | 1.17 |
| | + Goals-Only | 88.8 | 89.1 | **1.61** | 18.9 | 0.54 | 1.71 |
| | + Goals-and-Constraints | 87.4 | 88.1 | 1.59 | 17.5 | **0.47** | 0.93 |
| | + Spatial Prompt **(ours)** | **90.7** | **91.3** | 1.40 | **7.7** | 0.55 | **0.64** |
| | 3-Body Problem **(ours)** | 85.9 | 87.0 | 1.51 | 11.9 | 0.56 | 1.12 |

Table 1: Results on HD-EPIC. Notice that **bold** is across the various prompts of Gemini2.5. Underline implies best metric per column. *Qwen2.5-VL-72B could only generate results on a subset of 51/80 videos and fails for the rest. [†] uses action segment *ground-truth*.

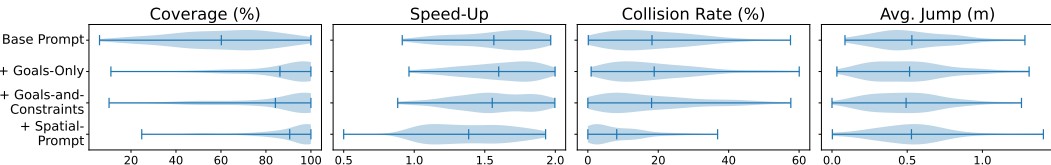

Figure 2: Distribution of evaluation metrics across all 100 videos as Gemini2.5 prompt is elaborated.

## 4 RELATED WORKS

**Egocentric Videos**. Popularised by recent datasets Grauman et al. (2022); Damen et al. (2022); Grauman et al. (2024); Huang et al. (2024); Perrett et al. (2025), egocentric vision has produced powerful models for parsing human activity streams. Current research span from action understanding Huang et al. (2020); Plizzari et al. (2025); Girdhar & Grauman (2021), multimodal sensing Huang et al. (2018); Kazakos et al. (2019); Zhang et al. (2024), to vision-language tasks Lin et al. (2022); Xu et al. (2024); Zhou et al. (2025); Kang et al. (2025). While this paradigm excels at passive understanding along one timeline, it does not address the challenge we study: how to re-allocate the same work across multiple concurrent workers while remaining faithful to real-world feasibility (space, objects and semantics). Task-graph annotation Mao et al. (2023); Grauman et al. (2024) is a promising bridge for enforcing causal order. However, these annotations are not publicly available, limiting comprehensive multi-constraint evaluation. We here focus on predicting parallel execution with goals and constraints that leverage readily available signals.

**Task Parallelisation**. Prior works on multitasking and coordination shows that human activity often interleaves goals and that latent "threads" can be unwoven from a single stream, revealing headroom for concurrency Price et al. (2022); Shen & Elhamifar (2025). Multi-person egocentric and mixed-view collections further indicate that collaborative behaviours are learnable in principle Jia et al. (2020; 2022); Xu et al. (2025); Ringe et al. (2025). However, prior works focus only on describing or segmenting concurrent intentions. Jia et al. (2020) is the closest work to our paper, as the Lemma dataset records one person performing the task, then records two people collaborating to complete the same task without verbal communication. While the intuition matches others, the videos are short-term (only 2 mins). Additionally, none of the works above predicts a parallel execution from a single demonstration. At the abstract level, our problem is analogous to task scheduling in parallel computing Dutot et al. (2004); Stern (2019): given a set of dependent tasks and limited processors, the goal is to minimise the *makespan*. Since optimal scheduling is difficult, practical solutions rely on heuristics (*e.g.* list scheduling such as HEFT) that deliver approximations efficiently Topcuoglu et al. (2002). We draw on this by including a HEFT-style baseline adapted to our problem.

**Spatial Reasoning in Vision-Language Models**. While Vision-Language Models (VLMs) have demonstrated impressive capabilities, recent work has consistently shown that spatial reasoning remains a significant weakness Stogiannidis et al. (2025); Wang et al. (2024a); He et al. (2025). This limitation is often attributed to the lack of explicit 3D and geometric information in pre-training. Current research aims to mitigate this through methods like fine-tuning on synthetic 3D VQA datasets Liu et al. (2025); Zha et al. (2025); Ogezi & Shi (2025) or developing specialized architectures Chen et al. (2024); Marsili et al. (2025); Song et al. (2025). We address this limitation from a task-driven angle. Rather than pursuing a universally capable spatial reasoner, we ask

| | Method | Coverage (%)↑ | Action Cov. (%) ↑ | Speed-Up ↑ | Coll. Rate (%) ↓ | Avg. Jump (m) ↓ | OCR (%) ↓ |
|---|---|---|---|---|---|---|---|
| 2-Body Problem | Naive Half-Half | 100.0 | 100.0 | 2.00 | 29.1 | 0.00 | - |
| | HEFT 1-min | 99.5 | 100.0 | 1.96 | 44.2 | 0.63 | - |
| | HEFT GT start-end† | 73.9 | 98.5 | 1.70 | 40.1 | 0.36 | - |
| | Qwen2.5-VL-72B* | 41.0 | 37.8 | 0.89 | 0.3 | 0.12 | - |
| | Gemini2.5 (Base Prompt) | 55.2 | 55.3 | 1.52 | 22.3 | 0.52 | - |
| | + Goals-Only | 75.8 | 76.9 | 1.54 | 18.8 | 0.44 | - |
| | + Goals-and-Constraints | 80.1 | 80.0 | **1.57** | 21.3 | **0.37** | - |
| | + Spatial Prompt **(ours)** | **89.9** | **90.6** | 1.35 | **10.1** | 0.43 | - |
| 3-Body Problem **(ours)** | | 83.0 | 84.9 | 1.64 | 18.2 | 0.51 | - |

Table 2: Results on EPIC. Notice that **bold** is across the various prompts of Gemini2.5. Underline implies best metric per column. *Qwen2.5-VL-72B could only generate results on a subset of 7/20 videos. † uses *ground-truth* action segments.

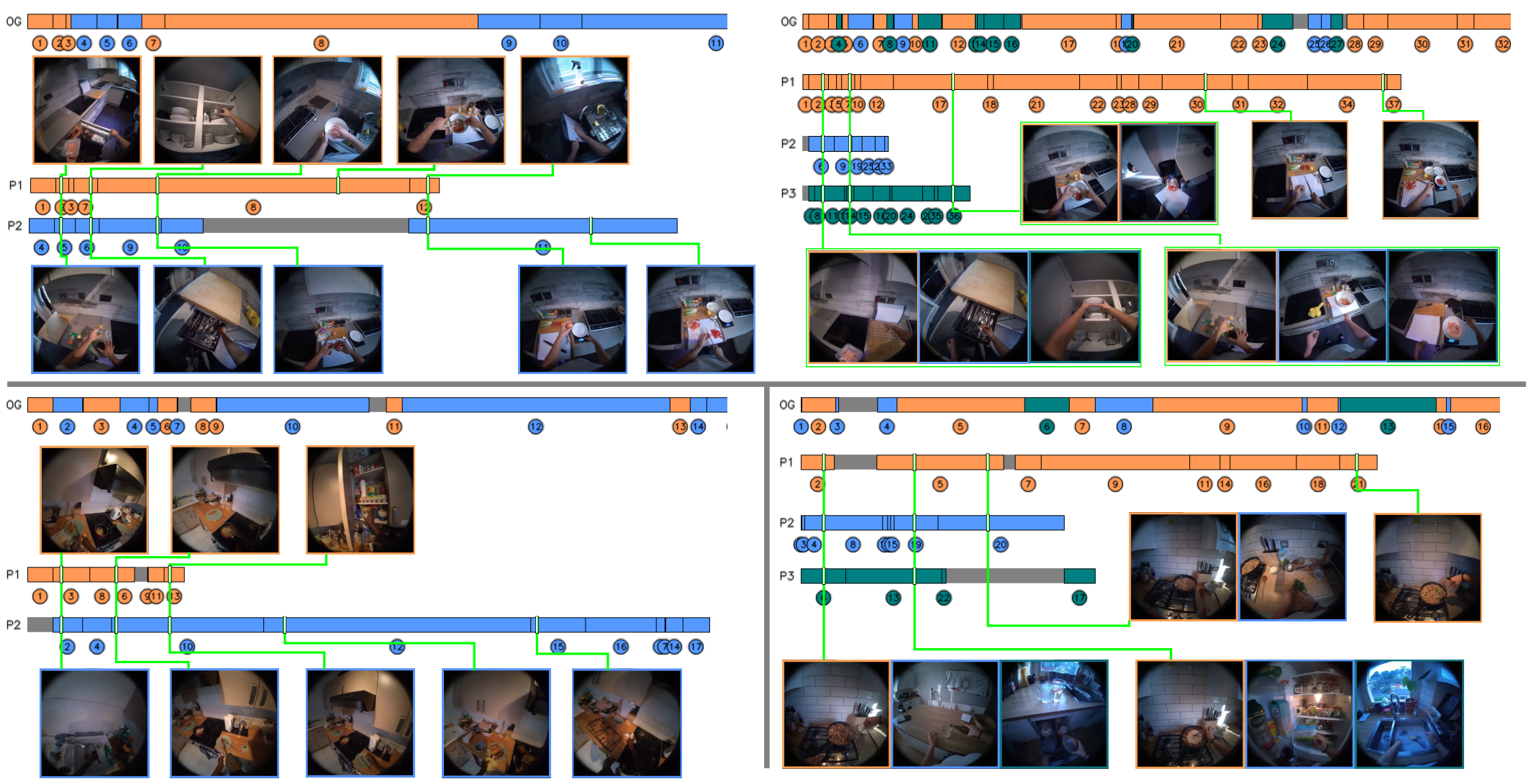

Figure 3: Qualitative results. **Top**: Same video with $N = 2$ and $N = 3$. Left: P1 is marinating chicken and washing up. P2 gathers spices (for the marination) in advance then prepares eggs. Right: P2 mostly brings items around. **Bottom Left**: $(N = 2)$ P2 is left to do the washing up and clearing. **Bottom Right**: $(N = 3)$ P1 is cooking, P2 is pouring a glass of wine then puts stuff away. P3 is emptying the dishwasher then contributes to washing up.

how to elicit reliable spatial behaviour for our downstream goal: collision avoidance in predicting parallel execution. Our spatial prompt discretises the environment and break the temporal timeline into zone-aligned periods in order to expose agent occupancy to the VLM and guide its reasoning without changing model weights.

## 5 Conclusion and Limitations

In this work, we introduce the N-Body Problem: predicting N-body parallel execution from a single egocentric video. We design a structured prompt that guides a Vision-Language Model (VLM) to reason about the spatial, object and causal constraints, predicting the parallel execution, maximising goals while respecting constraints. Tested on 100 videos, our experiments show the difficulty of the N-Body Problem, and that Gemini 2.5 Pro can be guided by dedicated prompts to produce execution with high coverage and reasonable speed-up while maintaining low constraint violations.

While guided Gemini 2.5 Pro achieves remarkable results, we identify some of its limitations. For example, the model fails to maintain the exact time when needed – *e.g.*, the time where food should be boiling/brewing or in the microwave/oven is not maintained. Some task dependencies cannot be automatically discovered – *e.g.*, grinding coffee beans is not automatically understood as a pre-requisite to brewing coffee. It also relies heavily on our explicit spatial prompt to avoid spatial collisions. These challenges suggest avenues for future research, to better model causality for unseen activities or develop architectures with more robust intrinsic spatial reasoning capabilities.

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

# A  THE COMPLETE PROMPT

```
1  # Task Overview
2
3  You are a job planner for a two-agent system.
4  You watch an egocentric video.
5  There are opportunities to parallelise the video to be performed by two
       agents (P1 and P2) efficiently.
6  You need to find such opportunities and make a parallel execution out of
       the original video, while respecting all dependencies and avoiding
       spatial conflicts.
7  The parallel execution shouldn't have dependency disorder (e.g. pour milk
       before bottle is taken out of the fridge) and shouldn't have 3D
       spatial conflict (e.g. both accessing the fridge).
8  Tell me how to rearrange the clips from the original video into two
       streams, so that the video can be performed by two people jointly.
9
10 # Avoiding Spatial Conflict
11
12 The trajectory CSV file lists the segments where the original person
       stays in different zones: "Z1", "Z2", "Z3", etc.
13 If two tasks from the original video could semantically be done in
       parallel, but the CSV file shows they both occur in the same zone "
       Z_n", you *must* serialise them.
14 Schedule P1 to work only in "Z_i" durations and P2 to work only in "Z_j"
       zone such that Z_i != Z_j.
15 The spatial reasong should rely on the trajectory csv exclusively, do not
       use any knowledge from the video.
16 While splitting agents in different zones, remember to still respect the
       semantics of the task execution.
17
18 # Avoiding Semantic Dependency Disorder
19
20 Below are more examples of semantic dependency disorder:
21 1. When making coffee, pour hot water into the coffee before the coffee
       powder has been added to the mug.
22 2. Wash mixer's head while the other agent is still using the mixer.
23 3. Uses the same trash bin at the same time.
24
25 # Covering every moment in the original video
26
27 Note that although this task aims to speed up the video, it should not
       skip any part of the original video. Every second of the original
       video needs to be covered, ensuring coverage = 100%.
28
29 # Format
30
31 Output json format:
32 With two keys "P1" and "P2", each contains a list of jobs (dictionaries).
33 A job is a part of the original video performed by P1 orP2.
34 A job is a dictionary with fields:
35 "new_start": the new start time performed by this person, "start" and "
       end" are the old start/end time in the original video. "text" is the
       summary text describing the clip of the job.
36 Example:
37 ```
38 {
39     "P1": [
40         {
41             "new_start": "00:00",
42             "start": "00:00",
43             "end": "00:26",
44             "text": "walk in and take from fridge",
45         },
```

```
46          {
47              "new_start": "00:26",
48              "start": "00:51",
49              "end": "01:14",
50              "text": "use and put milk into fridge"
51          },
52          {
53              "new_start": "00:49",
54              "start": "01:22",
55              "end": "1:59",
56              "text": "take cereal and mix"
57          }
58      ],
59      "P2": [
60          {
61              "new_start": "00:00",
62              "start": "00:26",
63              "end": "00:44",
64              "text": "put cloth and move bottles",
65          },
66          {
67              "new_start": "00:19",
68              "start": "00:44",
69              "end": "00:51",
70              "text": "take out cup and put onto the table",
71          },
72          {
73              "new_start": "00:26",
74              "start": "01:14",
75              "end": "01:22",
76              "text": "prepare spoon",
77          }
78      ]
79 }
80 ```
81
82 # Fall-back to single-thread
83
84 When it is really hard to schedule more than one agent in a constrained
       spatial space, a safer solution with single agent is preferred than a
       dangerous parallel plan with high potential spatial collision.
85
86 # Requirement Summary
87
88 Main requirement: Coverage. To avoid missing any segments of the full
       video (coverage < 100%),
89 keep in mind that every segment in the new parallel plan corresponds to a
       segment in the original video, and the original segments add up
       should always cover 100% of the original duration.
90
91 Supplementary requirements:
92 1. Pay attention to Spatial Conflicts. For example, no simultaneous
       access to the fridge. A better planning may use the same countertop
       at different times.
93 2. Pay attention to the Object dependency, e.g. place mug before pouring
       milk.
```

| Method | Coverage (%) ↑ | Action Coverage (%) ↑ | Speed-Up ↑ | Collision Rate (%) ↓ | Avg. Jump (m) ↓ | OCR (%) ↓ |
|---|---|---|---|---|---|---|
| +Goals-and-Constraints | 87.4 | 88.1 | 1.59 | 17.5 | 0.47 | 0.93 |
| Raw Trajectory | 90.6 | 91.3 | 1.49 | 13.9 | 0.48 | 1.17 |
| GMM (5 comps) | 91.3 | 91.4 | 1.48 | 10.8 | 0.57 | 0.57 |
| GMM (10 comps) | 86.8 | 87 | 1.44 | 10.5 | 0.55 | 0.89 |
| $40 \times 40$cm | 88.3 | 88.8 | 1.48 | 11.5 | 0.48 | 0.90 |
| $80 \times 80$cm | 88.7 | 89.1 | 1.40 | 9.9 | 0.48 | 0.57 |
| $120 \times 120$cm | 90.7 | 91.3 | 1.40 | 7.7 | 0.55 | 0.64 |

Table 3: Spatial Prompt variants on HD-EPIC.

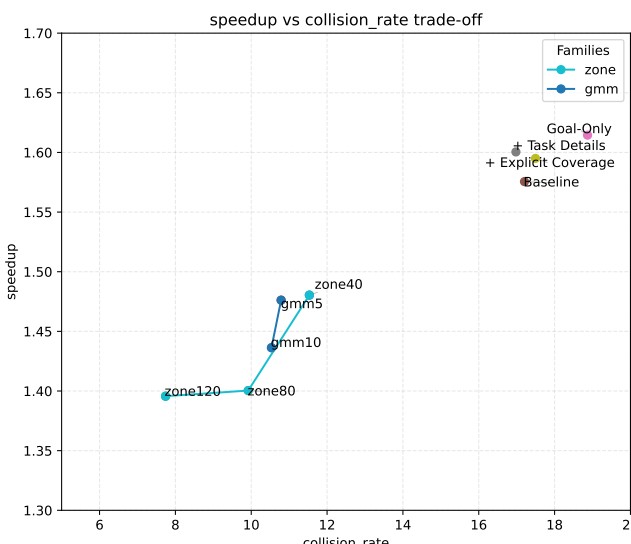

Figure 4: Speed-Up and Collision trade-off of different methods on the HD-EPIC dataset.

## B   SUPPLEMENTARY VIDEO

In the supplementary zip, we provide a folder for Fig 1 with the full video. This allows to compare the original input for our method (`original.mp4`[2]) to the output of **our** method for $N = 2$ (`N2.mp4`) and $N = 3$ (`N3.mp4`).

Additionally, we provide a diverse set of long clips from the various videos of HD-EPIC and EPIC-Kitchens in our demonstration video (`demo.MP4`). For $N = 2$, we have 3 videos from HD-EPIC and 2 videos from EPIC-Kitchens. For $N = 3$, we have 2 videos from HD-EPIC and 1 video from EPIC-kitchens. Note that all the videos are played at $2\times$ speed to fit the whole demonstration into 5min19s demonstration video.

## C   SPATIAL PROMPT VARIANTS

Above shown in Table 1 and Table 2, Gemini 2.5 Pro struggles with spatial collision of agents before introducing spatial prompt. In addition to the zone-related temporal segments introduced in 3.3, we experiment different variants of processing spatial trajectory: (i) with raw trajectory, (ii) equal-sized zones with different sizes and (iii) using Gaussian Mixture Model (GMM) to cluster trajectory into zone with dynamic sizes. The complete prompt uses zone size of 120x120cm.

Table 3 show comparison results of HD-EPIC. Raw trajectory reduces the collision rate by a small amount at the speed-up of $1.48\times$. However, GMM with 5 components and equal-sized zone of $40 \times 40$cm achieve the same speed-up with much less collision rate. Increasing zone sizes lead to more reduced collision rates, but the speed-up increases accordingly. With a larger zone size, the

---

[2]Note that this video is heavily compressed to fit in size. For full resolution, please refer to the HD-EPIC dataset video: `P06-20240510-121619`

collision rate reduces more, but the speed-up also increases: an agent is unable to work when another agent owns a larger part of the space exclusively.

We show how different parameter and variants all contributes to the reduced collision rate, while trading-off speed-up differently, within and across grouping variant. HD-EPIC results in fig. 4.

