# OpenReview forum: "The N-Body Problem: Predicting Parallel Execution from Single-Person Egocentric Video"
_ICLR.cc/2026/Conference — ICLR 2026 Conference Withdrawn Submission_

### Official Review · Reviewer_YiaF · 2025-10-29

**Soundness:** 4
**Presentation:** 3
**Contribution:** 4
**Rating:** 4
**Confidence:** 3

**Summary:**

This paper proposes a novel framework for decomposing egocentric videos of single-agent tasks into parallelizable multi-agent execution plans, coined as the N-body problem for video understanding. Leveraging structured prompting with a Vision-Language Model (VLM), the authors demonstrate that collaborative execution plans (e.g., 2-body and 3-body) can be synthesized with consideration for physical, spatial, and causal constraints. Experimental evaluations on HD-EPIC and EPIC-KITCHENS datasets show that their prompting-based approach outperforms heuristic baselines and achieves better trade-offs between coverage, speed-up, and constraint violations.

**Strengths:**

* Novel Perspective: The core idea of transforming sequential task demonstrations into collaborative execution plans is both creative and timely, especially for robotics, home assistants, or collaborative AI systems.

* Scalable Prompting Strategy: The structured prompt engineering is cleverly designed and shows practical effectiveness without requiring model finetuning.

* Empirical Insight: The paper provides compelling qualitative and quantitative analysis, particularly the trade-off between speed-up and constraint violations as the number of agents increases (e.g., from 2-body to 3-body).

**Weaknesses:**

* No Explicit Modeling of Multi-Agent Orchestration: While the paper tackles multi-agent execution planning, it does so implicitly through VLM-driven segmentation and allocation. There is no explicit mechanism or learned model that handles orchestration dynamics such as role assignment.

* Evaluated Plans Are Static: The output plans are static and non-adaptive. In real-world multi-agent orchestration, plans often need to be robust to execution variance (e.g., one agent moving slower, object unavailability), which is not addressed in this work. That said, this limitation seems more aligned with future work directions rather than a fundamental flaw.

**Questions:**

* Agent-level Orchestration Modeling: Have you considered incorporating an explicit orchestration module (e.g., centralized or decentralized planner) to better model inter-agent dependencies and coordination, rather than relying entirely on the VLM's prompt-based reasoning?

* Plan Adaptability: The current output plans appear static and offline. Do you envision integrating online feedback mechanisms or real-time adaptation (e.g., replanning based on agent delays or failures)?

While the proposed approach introduces several valuable and creative design choices, some parts of the system remain unclear to me. I outline a few questions below to better understand the method's assumptions and practical implications.

**Details Of Ethics Concerns:**

None.

---

### Official Review · Reviewer_DGvi · 2025-10-31

**Soundness:** 2
**Presentation:** 1
**Contribution:** 1
**Rating:** 2
**Confidence:** 4

**Summary:**

This paper introduces a new task, the N-Body Problem, which aims to infer multi-agent parallel execution plans from single-person egocentric videos. The authors propose using a commercial vision-language model (Gemini 2.5 Pro) guided by structured prompts to assign task segments to hypothetical agents under spatial, object, and causal constraints.
The paper evaluates this setup using pseudo-metrics (coverage, speed-up, collision rate, etc.) on HD-EPIC and EPIC-Kitchens datasets, showing that structured prompting improves these numbers over baselines.

The idea is interesting in spirit — learning collaborative affordances from single-person data — but the current formulation and methodology raise significant concerns.

**Strengths:**

The high-level motivation (inferring potential collaboration from egocentric data) is original and thought-provoking.

**Weaknesses:**

- **The proposed “N-Body Problem” feels largely problem-driven rather than insight-driven.**
In my view, the more fundamental and meaningful research problem lies in robustly extracting and representing a task graph - one that faithfully captures the temporal, spatial, and object-level dependencies among subtasks in egocentric video - rather than in defining the so-called “N-Body Problem.”
If such a task graph were reliably constructed (with each node representing a subtask and edges encoding temporal order, resource constraints, and spatial exclusivity), then the downstream scheduling of N agents becomes a deterministic problem. Classical methods in multi-processor scheduling, constraint optimization, or resource-constrained project scheduling (RCPSP) can already solve it efficiently and transparently.
In that sense, the “N-Body Problem” as introduced here seems to be a problem created for the sake of having a new formulation. It bypasses the more substantive challenge - learning or inferring the task graph itself - and instead focuses on generating pseudo-parallel plans from a single demonstration using heuristic prompting. The result feels less like addressing a fundamental research question, and more like constructing an artificial benchmark to justify prompt-based reasoning.

- **Lack of Ground Truth and Proper Evaluation**
For such a synthetic and unconventional setup, ground truth data are essential.
The paper repeatedly mentions that no multi-agent parallel executions exist, and therefore resorts to pseudo-metrics like “coverage” and “speed-up” computed from the same single-person videos.
This undermines the credibility of the evaluation:
  - The model is tested on its own synthetic metrics, not on real-world validation.
  - There is no behavioral, human, or even simulated ground truth to measure whether the generated plans are feasible or efficient.
  - Metrics like “speed-up” and “coverage” are uncalibrated and may not correlate with meaningful parallelism.

  If the authors wish to define a new problem, they should also define a minimal benchmark with genuine ground truth — e.g., recording the same tasks performed collaboratively by multiple humans — rather than relying on self-referential measures.

- The central “method” is essentially a handcrafted prompting template with additional CSV-based spatial context.
While this may yield qualitative improvements, it is not a novel algorithmic contribution and depends entirely on a closed-source proprietary model (Gemini 2.5 Pro).

**Questions:**

Please see weaknesses.

---

### Official Review · Reviewer_km5W · 2025-11-01

**Soundness:** 3
**Presentation:** 2
**Contribution:** 2
**Rating:** 4
**Confidence:** 3

**Summary:**

This paper introduces and formalizes a new task called the N-Body Problem. The core question is: given a single egocentric video of one person performing a complex set of tasks, can a model generate a valid plan for N people (or agents) to perform the same set of tasks in parallel? The goal is to maximize speed-up and task coverage and respect 1. Spatial constraints; 2. Object constraints; 3. Causality constraints. To solve this, they propose a structured prompting strategy to guide a state-of-the-art Vision-Language Model (VLM), Gemini 2.5 Pro. The key to their method is providing the VLM with explicit spatial and temporal information. They discretize the 3D environment into "zones" on the ground plane and provide the VLM with a CSV file detailing the time segments the original agent spent in each zone. The prompt then explicitly instructs the VLM to avoid scheduling two agents in the same zone at the same time. The authors evaluate their method on 100 long videos (avg. 25 min) from the EPIC-Kitchens and HD-EPIC datasets. Results show their full method significantly outperforms baselines.

**Strengths:**

1. The “N-body problem” is novel and well-defined task that pushes video understanding from passive observation to active, generative reasoning and planning. This has clear implications for robotics, simulation, and embodied AI.
2. The formalization of this paper is a key strength. The clear and distinct separation of **Goals** (to maximize) from **Constraints** (to minimize)  creates a robust framework that is highly valuable for structuring future research in this area.

**Weaknesses:**

1. While the paper introduces a novel and compelling "N-Body Problem," its current solution relies on a set of idealized assumptions and "privileged" information. As you've noted, these strong prerequisites significantly limit the method's applicability and feasibility in real-world scenarios.
    1. Dependency on Ground-Truth Camera Poses: This is the most significant limitation. The method's ability to achieve low collision rates is almost entirely dependent on the Spatial Prompt, which is generated from ground-truth camera poses. In any practical application (e.g., a robot learning from a web video), this perfect, noise-free 3D trajectory is unavailable.
    2. Simplistic "No Extra Objects" Assumption: The work evaluates object conflicts using an unrealistic "closed-world" assumption. It assumes that only the specific *instances* of objects used in the video exist (e.g., if one knife is used, the model assumes only one knife is available in the entire kitchen) in L195. This is fundamentally not how real-world environments work, as kitchens often contain multiple instances of common tools (spoons, bowls, knives).
    3. Rigid and Unrealistic Handling of "Wait" Times: As you pointed out in L200, the model's handling of "process" tasks (like an oven baking, water boiling, or a microwave running) is deeply flawed and inconsistent with human behavior.
2. Lack of Technical Contribution and Reliance on a Non-Deterministic "Black Box".
    1. Minimal Technical Novelty: The paper's core method is not a new algorithm, model architecture, or learning technique. It is a "prompt engineering" solution that relies entirely on the pre-existing, proprietary capabilities of Gemini 2.5 Pro. While defining the problem and metrics is a contribution, the *solution* itself lacks technical depth and novelty.
    2. No Guarantees of Optimality or Correctness: The solution is non-deterministic and lacks formal guarantees. As you rightly noted, VLMs are prone to hallucination and logical errors. There is no way to *ensure* the VLM's generated plan is correct, let alone optimal. The problem of task scheduling is a classic (NP-hard) optimization problem, yet this method replaces verifiable algorithms with a probabilistic, "black-box" heuristic that may fail in subtle or unpredictable ways (e.g., missing a less-obvious causal dependency) without any recourse.
3. For the N-body problem for a given video, how do you determine what the optimal solution is? It seems there could be multiple solutions; how would you evaluate the diversity of these solutions?

**Questions:**

My main concerns focus on the problem definition, reliance on VLMs and the evaluation of the method. I acknowledge the contribution of the paper for proposing the “N-body” problem, yet I believe that the current version is inadequate for practical applications.

---

### Official Review · Reviewer_br3n · 2025-11-08

**Soundness:** 2
**Presentation:** 3
**Contribution:** 2
**Rating:** 4
**Confidence:** 4

**Summary:**

This paper introduces a novel task called the N-Body task, which explores how N individuals can collaboratively perform the same set of actions observed in a given video. The authors claim that the primary goal of this task is to maximize speed-up. To tackle this problem, the authors first propose a set of metrics to evaluate both performance and feasibility. Additionally, they introduce a structured prompting strategy to guide Vision-Language Models (VLMs) in generating viable parallel executions. The authors validate the effectiveness of their proposed method through experiments on the EPIC-Kitchens and HD-EPIC datasets.

**Strengths:**

- The writing and structure of the paper are clear and easy to follow.
- The proposed N-Body Problem is intriguing and likely has real-world applications.
- The proposed evaluation metrics appear reasonable and capable of effectively measuring the goals and constraints.

**Weaknesses:**

- Although the authors propose a structured prompting strategy to guide VLMs in addressing this problem, I believe it is unnecessary to have the VLM directly infer the final task execution results. A more reasonable approach would be to use VLMs and expert models to identify sub-actions and establish causal relationships between them, followed by traditional optimization algorithms to derive the final solution. I do not think that relying on prompt engineering to have the VLM solve this in one step offers practical value.
- This work lacks substantial technical contributions. It primarily introduces a problem and devotes significant space to defining and describing evaluation metrics, while the proposed prompt design lacks technical innovation. This resembles a technical report rather than an academic paper.
- The authors only conduct experiments on a single VLM. To demonstrate the effectiveness of the method, I believe it is necessary to test the proposed approach on more models and additional tasks.

**Questions:**

See Weaknesses

---

### Note · Authors · 2025-11-12

I have read and agree with the venue's withdrawal policy on behalf of myself and my co-authors.